# Searching for Fairer Machine Learning Ensembles

**Michael Feffer**[1]  **Martin Hirzel**[2]  **Samuel C. Hoffman**[2]
**Kiran Kate**[2]  **Parikshit Ram**[2]  **Avraham Shinnar**[2]

[1]Carnegie Mellon University, Pittsburgh, PA, USA
[2]IBM Research, Yorktown Heights, NY, USA

**Abstract**  Bias mitigators can improve algorithmic fairness in machine learning models, but their effect on fairness is often not stable across data splits. A popular approach to train more stable models is ensemble learning, but unfortunately, it is unclear how to combine ensembles with mitigators to best navigate trade-offs between fairness and predictive performance. To that end, we extended the open-source library Lale to enable the modular composition of 8 mitigators, 4 ensembles, and their corresponding hyperparameters, and we empirically explored the space of configurations on 13 datasets. We distilled our insights from this exploration in the form of a guidance diagram that can serve as a starting point for practitioners that we demonstrate is robust and reproducible. We also ran automatic combined algorithm selection and hyperparmeter tuning (or CASH) over ensembles with mitigators. The solutions from the guidance diagram perform similar to those from CASH on many datasets.

## 1 Introduction

Algorithmic bias in machine learning can lead to models that discriminate against underprivileged groups in various domains, including hiring, healthcare, finance, criminal justice, education, and even child care. Of course, bias in machine learning is a socio-technical problem that cannot be solved with technical solutions alone. That said, to make tangible progress, this paper focuses on *bias mitigators*, which improve or replace an existing machine learning estimator (e.g., a classifier) so it makes less biased predictions (e.g., class labels) as measured by a fairness metric (e.g., disparate impact [16]). Unfortunately, bias mitigation often suffers from high *volatility*, meaning the estimator is less stable with respect to group fairness metrics. In the worst case, this volatility can even cause a model to appear fair when measured on training data while being unfair on production data. Given that ensembles (e.g., bagging or boosting) can improve stability for accuracy metrics [38], we felt it was important to explore whether they also improve stability for group fairness metrics.

Unfortunately, the sheer number of ways in which ensembles and mitigators can be combined and configured with base estimators and hyperparameters presents a dilemma. On the one hand, the diversity of the space increases the chances of it containing at least one combination with satisfactory fairness and/or predictive performance for the provided data. On the other hand, finding this combination via brute-force exploration may be untenable if resources are limited.

To this end, we conducted experiments that navigated this space with 8 bias mitigators from AIF360 [7]; bagging, boosting, voting, and stacking ensembles from the popular scikit-learn library [11]; and 13 datasets of various sizes and baseline fairness (more than prior algorithmic fairness papers). Specifically, we searched the Cartesian product of datasets, mitigators, ensembles, and hyperparameters both via brute-force and via Hyperopt [8] for configurations that optimized fairness while maintaining decent predictive performance and vice-versa. Our findings confirm the intuition that ensembles often improve stability of both accuracy and group fairness metrics. However, the best configuration of mitigator and ensemble depends on dataset characteristics, evaluation metric of choice, and even worldview [18]. Therefore, we automatically distilled a method selection guidance diagram in accordance with the results from both brute-force search and Hyperopt exploration.

To support these experiments, we assembled a library of pluggable ensembles, bias mitigators, and fairness datasets. While we reused popular and well-established open-source technologies, we made several new adaptations in our library to get components to work well together. Our library is open-source (https://github.com/IBM/lale) to encourage research and real-world adoption.

## 2 Related Work

Some prior work used ensembles for fairness, but they used specialized ensembles and bias mitigators, whereas our work uses off-the-shelf modular components. The *discrimination-aware ensemble* uses a heterogeneous collection of base estimators [26]; when they all agree, it returns the consensus prediction, otherwise, it classifies instances as positive iff they belong to the unprivileged group. The *random ensemble* also uses a heterogeneous collection of base estimators, and picks one of them at random to make a prediction [21]. The paper offers a synthetic case where the ensemble is more fair and more accurate than all base estimators, but lacks experiments with real datasets. *Exponentiated gradient reduction* trains a sequence of base estimators using a game theoretic model where one player seeks to maximize fairness violations by the estimators so far and the other player seeks to build a fairer next estimator [1]. In the end, for predictions, it uses weights to pick a random base estimator. *Fair AdaBoost* modifies boosting to boost not for accuracy but for individual fairness [9]. In the end, for predictions, it gives a base estimator higher weight if it was fair on more instances from the training set. The *fair voting ensemble* uses a heterogeneous collection of base estimators [29]. Each prediction votes among base estimators $\phi_t$, $t \in 1..n$, with weights $W_t = \alpha \cdot A_t / (\Sigma_{t=1}^n A_j) + (1 - \alpha) \cdot F_t / (\Sigma_{t=1}^n F_j)$, with $A_t$ an accuracy metric and $F_t$ a fairness metric. The *fair double ensemble* uses stacked predictors, with a final linear estimator, with a novel approach to train the weights of the final estimator to satisfy a system of accuracy and fairness constraints [31].

Each of the above-listed approaches used an ensemble-specific bias mitigator, whereas we experiment with eight different off-the-shelf modular mitigators. Moreover, each of these approaches used one specific kind of ensemble, whereas we experiment with off-the-shelf modular implementations of bagging, boosting, voting, and stacking. Using off-the-shelf mitigators and ensembles facilitates plug-and-play between the best available independently-developed implementations. Unlike these earlier papers, our paper specifically explores fairness stability and the best ways to combine mitigators and ensembles. We auto-generate a guidance diagram from this exploration.

We are not the first to use automated machine learning, including Bayesian optimizers, to optimize models and mitigators for fairness [32, 39]. And it is widely accepted that ensembling is a critical part of AutoML (see for example auto-sklearn [17] and AutoGluon [14]). But unlike prior work, we focus on applying AutoML to ensemble learning and bias mitigation to validate our guidance diagram and results.

There are previous empirical studies of fairness techniques [10, 19, 20, 24, 30, 35, 36, 40]. However, only one explores fairness with ensembles [20], and it does not consider bias mitigators.

Our work also offers a new library of bias mitigators. While there have been excellent prior fairness toolkits such as ThemisML [4], AIF360 [7], and FairLearn [1], none support ensembles. Ours is the first that is modular enough to investigate a large space of unexplored mitigator-ensemble combinations. We previously published some aspects of our library in a non-archival workshop with no official proceedings, but did not discuss ensembles [23]. In another non-archival workshop paper, we discussed ensembles and some of these experimental results [15], but no Hyperopt results and only limited analysis of the guidance diagram. Such results and further analysis are included here. After collecting 13 fairness datasets for this paper, we collected 7 more, bringing the total to 20 [22].

## 3 Library and Datasets

One of our contributions is compatibility between mitigators from AIF360 [7] and ensembles from scikit-learn [11]. To provide the glue and facilitate searching over a space of mitigator and ensemble configurations, we extended the Lale open-source library for semi-automated data science [5, 6].

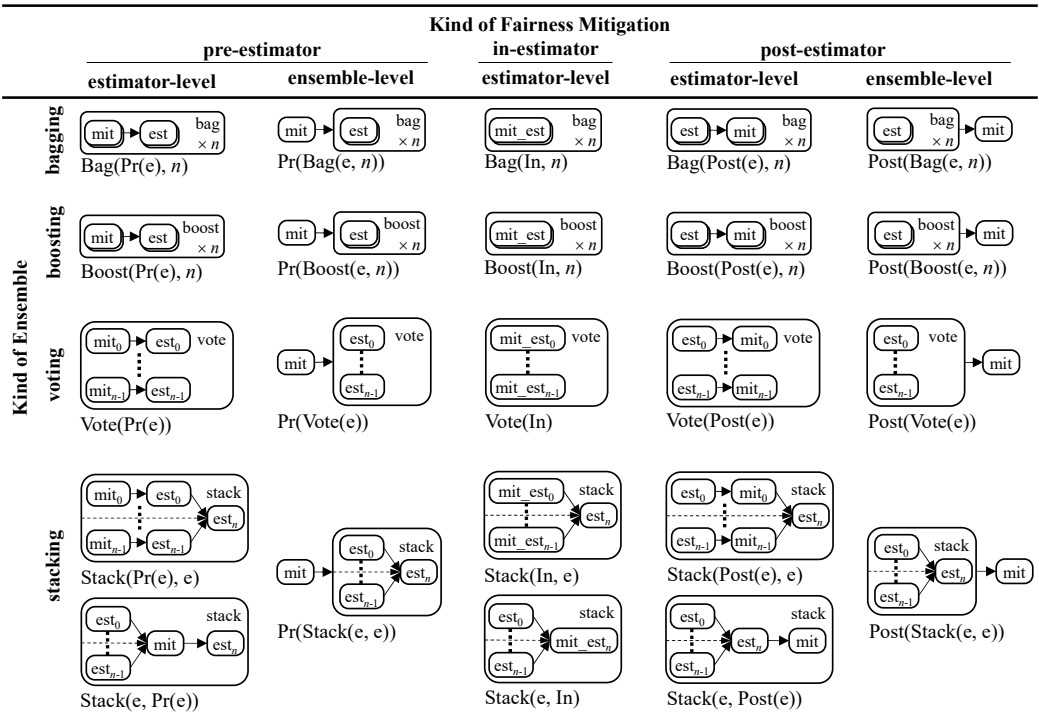

Figure 1: Combinations of ensembles and mitigators. Pr(e) applies a pre-estimator mitigator before an estimator e; In denotes an in-estimator mitigator, which is itself an estimator; and Post(e) applies a post-estimator mitigator after an estimator e. Bag(e, n) is BaggingClassifier with n instances of estimator e; Boost(e, n) is AdaBoostClassifier with n instances of e; Vote(e) is VotingClassifier with a list of estimators e; and Stack(e, e) is StackingClassifier with a list of estimators (first e) and a final estimator (second e). For stacking, the passthrough option is shown by a dashed arrow.

**Metrics.** This paper uses metrics from scikit-learn, including precision, recall, and $F_1$ score (harmonic mean of precision and recall). In addition, we implemented a scikit-learn compatible API for several fairness metrics from AIF360 including disparate impact (the ratio of positive outcomes for the unprivileged group versus those for the privileged group as described by Feldman et al. [16]). We also measure time (in seconds) and memory (in MB) utilized when fitting models.

**Ensembles.** Ensemble learning uses multiple weak models to form one strong model. Our experiments use four ensembles supported by scikit-learn: bagging, boosting, voting, and stacking. Following scikit-learn, we use the following terminology to characterize ensembles: A *base estimator* is an estimator that serves as a building block for the ensemble. An ensemble supports one of two *composition* types: whether the ensemble consists of identical base estimators (*homogeneous*, e.g. bagging and boosting) or different ones (*heterogeneous*, e.g. voting and stacking). Similarly, each ensemble supports one of two *training styles*: whether the ensemble trains base estimators one at a time sequentially (*series*, e.g. boosting) or independently from each other (*parallel*, e.g. bagging, voting, and stacking). For the homogeneous ensembles, we used their most common base estimator in practice: the decision-tree classifier. For the heterogeneous ensembles (voting and stacking), we used a set of typical base estimators: XGBoost [13], random forest, k-nearest neighbors, and support vector machines. Finally, for stacking, we also used XGBoost as the final estimator.

**Mitigators.** We added support in Lale for bias mitigation from AIF360 [7]. AIF360 distinguishes three kinds of mitigators for improving group fairness: *pre-estimator mitigators*, which are learned input manipulations that reduce bias in the data sent to downstream estimators (we used DisparateIm-

Table 1: Qualitative and quantitative summary information of the datasets. The datasets are ordered by first partitioning by whether they contain at least 8,000 rows (we picked 8,000 to get a roughly even split; the partition is represented by the horizontal line in the middle of the table) and then sorting by descending baseline disparate impact (DI). Values for feature importance ranking of most predictive protected attribute according to XGBoost (Importance), the number of columns ($N_{cols}$), number of rows ($N_{rows}$), and baseline disparate impact (DI) displayed here are computed *after* preprocessing techniques are applied.

| Dataset | Description | Privileged group(s) | Imp-ortance | $N_{cols}$ | $N_{rows}$ | DI |
|---|---|---|---|---|---|---|
| Compas violent | Correctional offender violent recidivism | White women | 4 | 10 | 3,377 | 0.822 |
| Credit-g | German bank data quantifying credit risk | Men and older people | 22 | 58 | 1,000 | 0.748 |
| Compas | Correctional offender recidivism | White women | 5 | 10 | 5,278 | 0.687 |
| Ricci | Fire department promotion exam results | White men | 6 | 6 | 118 | 0.498 |
| TAE | University teaching assistant evaluation | Native English speakers | 1 | 6 | 151 | 0.449 |
| Titanic | Survivorship of Titanic passengers | Women and children | 2 | 37 | 1,309 | 0.263 |
| SpeedDating | Speed dating experiment at business school | Same race | 24 | 70 | 8,378 | 0.853 |
| Bank | Portuguese bank subscription predictions | Older people | 17 | 51 | 45,211 | 0.840 |
| MEPS 19 | Utilization results from Panel 19 of MEPS | White individuals | 22 | 138 | 15,830 | 0.490 |
| MEPS 20 | Same as MEPS 19 except for Panel 20 | White individuals | 18 | 138 | 17,570 | 0.488 |
| Nursery | Slovenian nursery school application results | "Pretentious parents" | 3 | 25 | 12,960 | 0.461 |
| MEPS 21 | Same as MEPS 19 except for Panel 21 | White individuals | 10 | 138 | 15,675 | 0.451 |
| Adult | 1994 US Census salary data | White men | 19 | 100 | 48,842 | 0.277 |

pactRemover [16], LFR [41], and Reweighing [25]); *in-estimator mitigators*, which are specialized estimators that directly incorporate debiasing into their training (AdversarialDebiasing [42], GerryFairClassifier [28], MetaFairClassifier [12], and PrejudiceRemover [27]); and *post-estimator mitigators*, which reduce bias in predictions made by an upstream estimator (we used CalibratedEqOddsPostprocessing [33]).

Fig. 1 visualizes the combinations of ensemble and mitigator kinds we explored, while also highlighting the modularity of our approach. Mitigation strategies can be applied at the level of either the base estimator or the entire ensemble, although not all combinations are feasible.

First, post-estimator mitigators typically do not support `predict_proba` functionality required for some ensemble methods and recommended for others. Calibrating probabilities from post-estimator mitigators has been shown to be tricky [33], so despite Lale support for other post-estimator mitigators, our experiments only explored CalibratedEqOddsPostprocessing.

Additionally, it is impossible to apply an in-estimator mitigator at the ensemble level, so we exclude those combinations. Finally, we decided to omit some combinations that are technically feasible but less interesting. For example, while our library supports mitigation at multiple points, say, at both the ensemble and estimator level of bagging, we elided these configuration from Fig. 1 and from our experiments.

**Datasets.** We gathered the datasets for our experiments primarily from OpenML [37]; the exceptions come from Medical Expenditures Panel Survey (MEPS) data [2, 3] and ProPublica data [34] not hosted there. Some have been used extensively as benchmarks elsewhere in the algorithmic fairness literature. We pulled other novel datasets from OpenML that have demographic data that could be considered protected attributes (such as race, age, or gender) and contained associated baseline levels of disparate impact. In addition, to get a sense for the predictive power of each protected attribute, we fit XGBoost models to each dataset with five different seeds and found the ranking of the average feature importance (where 1 is the most important) of the most predictive protected attribute for that dataset. In all, we used 13 datasets, with most information summarized in Table 1 and granular feature importance information summarized in the Appendix. When running experiments, we split

the datasets using stratification by not just the target labels but also the protected attributes [23], leading to moderately more homogeneous fairness results across different splits. The exact details of the preprocessing are in the open-source code for our library for reproducibility. We hope that bundling these datasets and default preprocessing with our package, in addition to AIF360 and scikit-learn compatibility, will improve dataset quality going forward.

## 4 Methodology

Given our 13 datasets, 4 types of ensembles, 8 mitigators, and all relevant hyperparameters, we wanted to gain insights about the best ways to combine ensemble learning and bias mitigation in various problem contexts and data setups. We compared the results of searching over the Cartesian product of these settings in two ways: a manual grid search to determine optimal configurations for each dataset and an automated search via Bayesian optimization in Hyperopt [8].

### 4.1 Grid Search

We organize our grid search experiments into two steps: a preliminary search that finds the "best" mitigators without ensembles, and subsequent experiments using those mitigator configurations.

**First step.** It is difficult to define "best" (in an empirical sense) given different dimensions of performance and datasets. We first run grid searches over each dataset, exploring mitigators and their hyperparameters with basic decision-trees where needed. We run 5 trials of 3-fold cross validation for each configuration. For each dataset, we choose a "best" pre-, in-, and post-estimator mitigator and (1) filter configurations to ones with acceptable fairness, ($0.8 \leq$ mean disparate impact $\leq 1.25$); (2) filter remaining to ones with nontrivial precision; (3) filter remaining to ones with good predictive performance, defined as mean $F_1$ score (across 5 trials) greater than both the average and median of all mean $F_1$ scores; (4) finally, select the mitigator with maximum precision (for Compas, prioritizing true positives) or recall (other datasets, avoiding false negatives). Tables 12 and 13 in our Appendix list the chosen pre-estimator and in-estimator configurations (the only post-estimator configuration is CalibratedEqOddsPostprocessing).

**Second step.** Given the "best" mitigator configurations, this step explores the Cartesian product of ensembles and mitigators of Fig. 1 plus ensemble hyperparameters. For bagging and boosting, the only ensemble-level hyperparameter varied between configurations was the number of base estimators: $\{1, 10, 100\}$ for bagging and $\{1, 50, 500\}$ for boosting. Voting and stacking use lists of heterogeneous base estimators as hyperparameters. In our experiments, these lists contained either 4 mitigated or 4 unmitigated base estimators. For the in-estimator mitigation case these were {PrejudiceRemover, GerryFairClassifier, MetaFairClassifier, and AdversarialDebiasing}. Lastly, stacking also has a `passthrough` hyperparameter controlling whether dataset features were passed to the final estimator. If `passthrough` is set to `False`, it is impossible to mitigate the final estimator due to lack of dataset features; otherwise we mitigate either the base estimators or final estimator, but not both. The second step also uses 5 trials of 3-fold cross validation for each experiment, running on a computing cluster with Intel Xeon E5-2667 processors @ 3.30GHz. Every experiment configuration run was allotted 4 cores and 12 GB memory.

### 4.2 Hyperopt Search

We used Hyperopt to perform another model configuration search, this time in a single step guided by an objective that combined predictive performance and fairness. We defined a single search space that includes all ensembles and mitigators and their hyperparameters. Then, we defined the blended scorer in Fig. 2 (L7 measures symmetric disparate impact and $F_1$ score; L8 scales both of these based on ranges determined in L4–L5; L9–L10 amplifies low outcomes to encourage AutoML to avoid them; and L11 returns the arithmetic mean). Finally, we ran Lale's Hyperopt wrapper, passing the `blended_scorer` as the objective to maximize and setting timeouts of 10 minutes per trial and 20 hours total for each dataset, on the same cluster as for grid search.

```
1  def symm_di(model, X, y):  # symmetric disparate impact
2      di = di_scorer(model, X, y)
3      return di if di <= 1 else 1 / di
4  min_di, max_di = symm_di.score_data(X=X, y_pred=y), 1
5  min_f1, max_f1 = f1_scorer(dummy, X, y), f1_scorer(xgboost, X, y)
6  def blended_scorer(model, X, y):
7      di, f1 = symm_di(model, X, y), f1_scorer(model, X, y)
8      di, f1 = (di - min_di) / (max_di - min_di), (f1 - min_f1) / (max_f1 - min_f1)  # scale
9      if di < 0.66: di -= 0.66 - di  # amplify low DI outcomes so AutoML avoids them
10     if f1 < 0.66: f1 -= 0.66 - f1  # amplify low F1 outcomes so AutoML avoids them
11     return 0.5 * (di + f1)  # blend to joint objective
```

Figure 2: Blended objective for Hyperopt search.

Table 2: Standardized Disparate impact Outcome (DO) and Volatility (DV). DO, DV use different scales.

| | No Mit. | | Pre- | | In- | | Post- | |
|---|---|---|---|---|---|---|---|---|
| | DO | DV | DO | DV | DO | DV | DO | DV |
| No ensemble | **0.42** | 0.18 | 0.73 | 0.38 | **0.87** | 0.44 | **0.53** | 0.24 |
| Bagging | 0.31 | **0.08** | 0.54 | **0.19** | 0.80 | 0.28 | 0.44 | **0.08** |
| Boosting | 0.33 | 0.18 | **0.69** | 0.39 | **0.87** | **0.26** | 0.41 | 0.12 |
| Voting | 0.29 | 0.09 | 0.51 | 0.35 | 0.40 | 0.45 | 0.21 | 0.20 |
| Stacking | 0.39 | 0.19 | 0.61 | 0.27 | 0.44 | 0.39 | 0.50 | 0.27 |

# 5 Results

This section includes quantitative results of our two searches and qualitative guidance regarding future model development based on these results.

## 5.1 Grid Search Results

**Result preprocessing.** To facilitate cross-dataset comparisons, we applied the following procedure on a per-dataset basis for each metric of interest: (i) given all results, map all values to the same region of metric space around the point of optimality if needed (i.e. for disparate impact, we use the reciprocal of a value if it is larger than 1 for downstream calculations, but for $F_1$, no modification is needed), and (ii) min-max scale the mean and standard deviation of the metric of interest, separately. After doing this for all datasets, we group remaining results by mitigator kind and ensemble type, and average the scaled values over all datasets for each group. Given a metric $M$, we refer to the result of this procedure using mean values as "standardized $M$ outcome" and using standard deviation as "standardized $M$ volatility". The tables and figures that follow report values normalized as described.

**Do ensembles help with fairness?** Table 2 shows the disparate impact results. Mitigation almost always improved disparate impact outcomes, but ensemble learning generally incurred a slight penalty, while generally reducing disparate impact volatility. In some contexts, this increased stability may be preferred over better yet more unstable predictions.

Table 3: Standardized $F_1$ outcome (FO) and volatility (FV). FO, FV use different scales.

| | No Mit. | | Pre- | | In- | | Post- | |
|---|---|---|---|---|---|---|---|---|
| | FO | FV | FO | FV | FO | FV | FO | FV |
| No ensemble | 0.70 | 0.20 | 0.54 | 0.39 | 0.51 | 0.49 | 0.63 | 0.19 |
| Bagging | **0.93** | 0.13 | 0.50 | **0.19** | 0.61 | **0.11** | 0.65 | **0.13** |
| Boosting | 0.84 | 0.28 | 0.49 | 0.25 | 0.52 | 0.28 | 0.63 | **0.13** |
| Voting | 0.77 | **0.09** | 0.40 | 0.36 | 0.45 | 0.50 | 0.58 | 0.19 |
| Stacking | 0.83 | 0.26 | **0.56** | 0.50 | **0.67** | 0.59 | **0.66** | 0.27 |

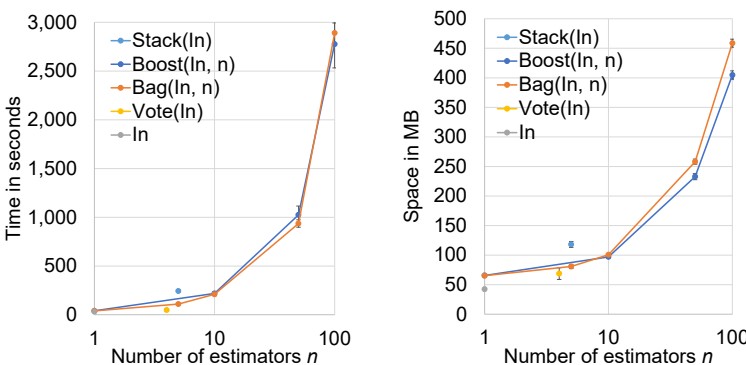

Figure 3: Resource consumption.

**Do ensembles help predictive performance when there is mitigation?** Table 3 shows $F_1$ results. Even with ensemble learning, mitigation decreases predictive performance, but relative to standalone mitigators, mitigated ensembles typically have better outcomes or stability, but not both. Except for a few cases, mitigated ensembles *can* help with predictive performance outcomes *or* volatility.

**How do ensembles affect resource consumption?** Fig. 3 reports the time and memory for training ensembles of in-estimator mitigators. We did not measure the overhead of the bias mitigators themselves, since it is determined by their implementation in AIF360 [7]. Time and memory are averaged over all datasets and all in-estimator mitigators in our experiments. Error bars reflect averages of standard deviations (each standard deviation calculated across all trial-folds for a given dataset and configuration). No min-max scaling was used to create this figure. Not surprisingly, more base estimators consume more resources, so we address this consideration in our guidance diagram.

### 5.2 Guidance for method selection

To advise future practitioners based on our results, we generated Fig. 4 from optimal configurations for particular metrics and data setups. To generate it, we do the following:

1. Organize all results by dataset.
2. Filter results for each dataset to ones that occur in the top 33% of results for both standardized disparate impact outcome and standardized $F_1$ outcome.
3. Place each result into one of four quadrants based on the dataset's baseline fairness and size.
4. Average each metric in each quadrant while grouping by model configuration.
5. Report the top 3 configurations per quadrant and metric.

**Leave-one-out evaluation.** One way in which we evaluate our guidance diagram is, for each dataset, to follow the diagram generation steps while leaving out the results pertaining to that dataset, and examine differences in terms of the recommended model configurations and their performances between the new diagram and the one generated from all of the datasets. Because our guidance diagram has three recommendations per metric, the largest number of differences between a leave-one-out diagram and the full dataset diagram for a given metric is three. We also compute signed differences of metric values by subtracting the metric value of the best model recommended by the leave-one-out diagram from that of the full dataset diagram. If the diagram creation method generalizes well, these differences should be close to zero. Table 4 displays both types of these differences for all omitted datasets and the metrics disparate impact mean, disparate impact standard deviation, and $F_1$ mean. Based on these differences, some datasets have more of an effect on the guidance diagram than others. This phenomenon will be covered in our discussion section.

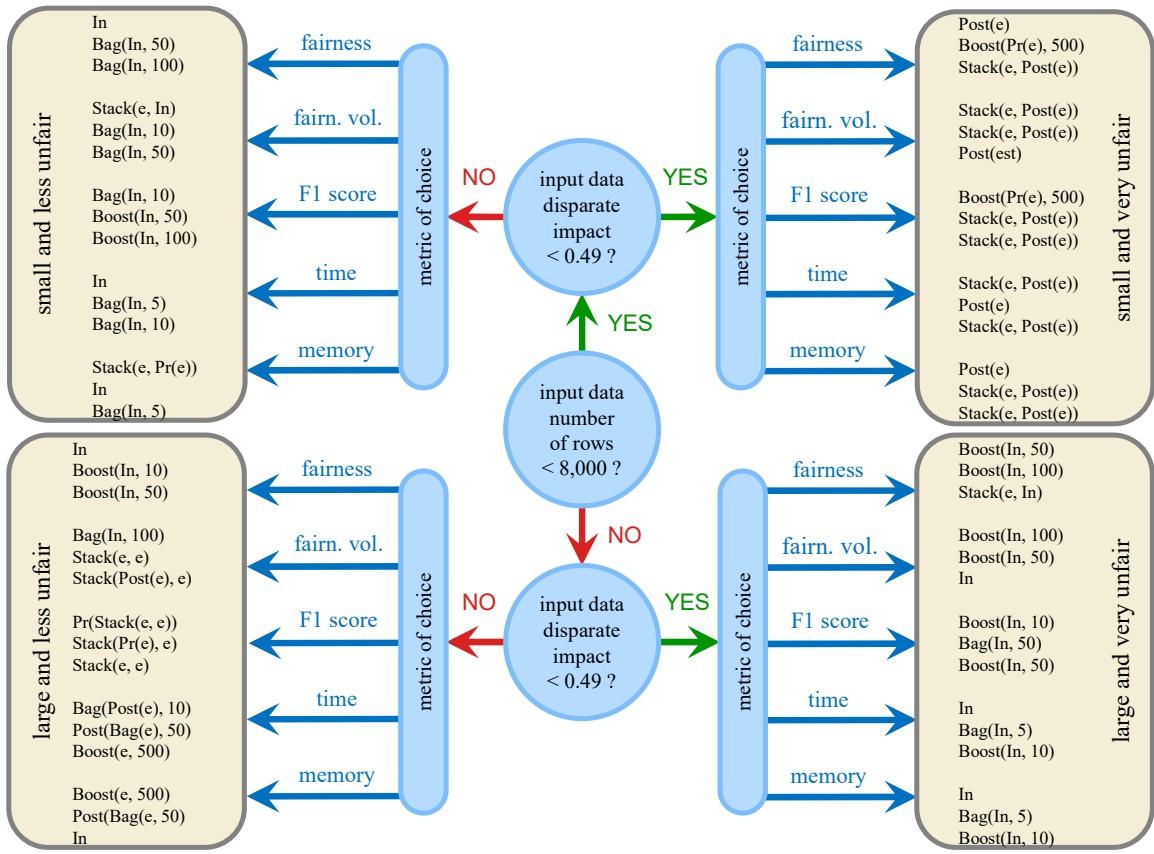

Figure 4: Guidance diagram for a good starting configuration given dataset properties and target metric.

### 5.3 Hyperopt Result Comparison

We purposely designed a scorer for Hyperopt (see methodology section) similar to the method we used to filter grid search results to produce the guidance diagram. Therefore, Hyperopt's solutions provide another way to evaluate the guidance diagram's suggested configurations.

Table 5 shows, for each dataset, the configurations returned by Hyperopt and recommended by the guidance diagram when average disparate impact or average $F_1$ score is the metric of interest. Fig. 5 shows the corresponding average $F_1$ score and disparate impact with standard deviations. A close inspection reveals that while the guidance diagram rarely recommends the exact same configuration as that found by Hyperopt, it often recommends one with similar performance.

## 6 Discussion

This section describes the impacts of our search results and guidance in addition to hypotheses informed by our results regarding biased data.

**Guidance diagram utility and robustness.** The previous section showed that the guidance diagram and Hyperopt search recommended configurations with relatively similar performance on most of the datasets. This suggests that the guidance diagram can recommend to practitioners starting points for model development based on their data setup and metric(s) of interest. Consulting the guidance diagram can be done quickly, without needing the time and compute resources of a search.

Our leave-one-out dataset experiments also suggest that our diagram generation algorithm is relatively robust to changes in the input data. This further supports the notion that our guidance diagram has useful recommendations. However, those experiments also showed that the presence or

Table 4: Number of configuration and signed metric differences between leave-one-out and full dataset guidance for omitted datasets. Note: metric differences are *not* standardized.

| | DI Mean | | DI StdDev | | F1 Mean | |
|---|---|---|---|---|---|---|
| Omitted Dataset | Num | Metric | Num | Metric | Num | Metric |
| COMPAS Violent | 0 | 0 | 0 | 0 | 0 | 0 |
| Credit-g | 3 | 0.29 | 3 | -0.03 | 3 | 0.23 |
| COMPAS | 0 | 0 | 0 | 0 | 0 | 0 |
| Ricci | 0 | 0 | 0 | 0 | 0 | 0 |
| TAE | 2 | 0.20 | 1 | 0 | 1 | -0.19 |
| Titanic | 1 | 0 | 1 | -0.12 | 1 | 0 |
| SpeedDating | 0 | 0 | 2 | 0 | 3 | 0.05 |
| Bank | 3 | 0.11 | 1 | -0.01 | 1 | 0 |
| MEPS 19 | 0 | 0 | 0 | 0 | 0 | 0 |
| MEPS 20 | 1 | 0.01 | 2 | 0 | 0 | 0 |
| Nursery | 0 | 0 | 0 | 0 | 0 | 0 |
| MEPS 21 | 1 | -0.04 | 1 | 0.03 | 1 | 0 |
| Adult | 0 | 0 | 0 | 0 | 0 | 0 |

Table 5: Configurations recommended by Hyperopt search and guidance diagrams optimized for fairness and predictive performance.

| Dataset | Hyperopt | Guidance F1 | Guidance DI |
|---|---|---|---|
| COMPAS V. | Pr(Stack(e, e)) | Bag(In, 10) | In |
| Credit-g | Vote(Pr(e)) | Bag(In, 10) | In |
| COMPAS | Bag(Post(e), 72) | Bag(In, 10) | In |
| Ricci | Pr(e) | Bag(In, 10) | In |
| TAE | In | Boost(Pr, 500) | Post(e) |
| Titanic | Pr(Vote(e)) | Boost(Pr, 500) | Post(e) |
| SpeedDating | In | Pr(Stack(e, e)) | In |
| Bank | Post(Boost(e, 206)) | Pr(Stack(e, e)) | In |
| MEPS 19 | In | Pr(Stack(e, e)) | In |
| MEPS 20 | Stack(e, In) | Boost(In, 10) | Boost(In, 50) |
| Nursery | Pr(e) | Boost(In, 10) | Boost(In, 50) |
| MEPS 21 | In | Boost(In, 10) | Boost(In, 50) |
| Adult | Pr(e) | Boost(In, 10) | Boost(In, 50) |

absence of certain datasets affected the resulting diagram more than others. For instance, the Credit-g and Bank datasets have more effects on the recommended configurations and model performance than the Adult or COMPAS datasets.

We attribute this phenomenon to the filtering of model results that takes place during diagram generation and properties of the datasets themselves. Most of the datasets in Table 1 that have large effects on the diagram have baseline disparate impact close to 0.8 (meaning they are relatively fair), and their protected attributes are not strongly predictive (based on feature importance ranking). This implies that with mitigation, it is possible to fit these datasets fairly and accurately. This in turn means model fitting results from those datasets comprise most of the results for the given quadrant after filtering to reasonable fairness and predictive performance. Therefore, when those datasets are missing, the generated diagram differs greatly from the one generated with all data. (The exceptions to this rule are TAE and Titanic. Given that those are the only two datasets in their quadrant and protected attributes are strongly predictive, it is difficult to fit either well in a fair manner. Therefore, neither contributes many fitting results after filtering, and both have tangible effects on the diagram.)

In light of how protected attribute feature importance of input datasets affects recommendations of the guidance diagram, one limitation of our diagram is its lack of branches for this property (thus

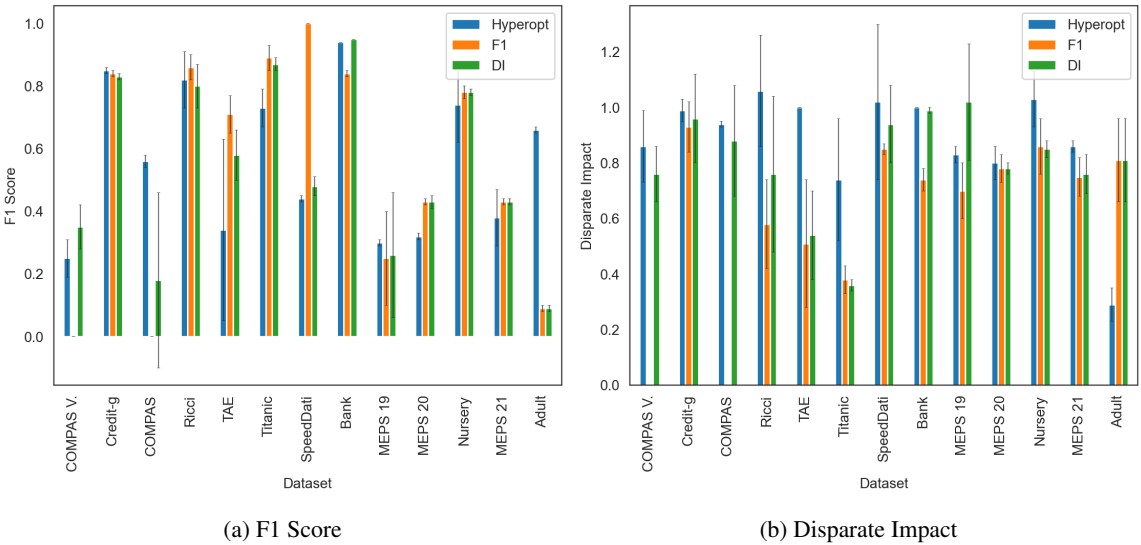

|  (a) F1 Score  |  (b) Disparate Impact  |

Figure 5: Average (with standard deviation) of F1 and DI for the recommended configurations (see Table 5) for each dataset from Hyperopt search, guidance diagram to optimize for predictive performance, and guidance diagram to optimize for fairness.

not providing recommendations based on this property). Determining this property requires training XGBoost models, which can take time and resources, while the other properties utilized can be quickly calculated. Thus, we still argue that our guidance diagram is useful to future practitioners.

**What is "good data?"** As mentioned by Holstein et al. [24], "future research" in the area of algorithmic fairness should "[develop] processes and tools for fairness-focused debugging" and "should also support practitioners in collecting and curating high-quality datasets in the first place". These recommendations suggest *how to collect good data?* and *what even* is *'good data'?* are questions with which the field is currently grappling.

We believe our results shed some light on these fronts. First, our findings suggest that converting "bad data" to "good data" may not (just) involve making datasets larger in number of *examples* but (also) making them larger in number of *features*. Prevailing notions of algorithmic fairness may imply that the best way to fix an unfair dataset is to add examples to reduce bias. While this may work, it could be difficult to do in practice (especially given societal mechanisms behind bias), and Holstein et al. [24] also raise that "How much data [one] would need to collect?" does not typically have a clear answer. However, our results imply that collecting more data to alleviate bias should be done by gathering more features instead of simply gathering more examples with the same features.

That being said, datasets like Adult and Ricci included attributes that were more predictive than protected attributes, yet they still did not strongly influence our guidance diagram. We conjecture that the more predictive attributes were highly correlated with protected attributes, and feature importance tables included in our Appendix seem to support this. Therefore, when collecting more features to reduce bias, one needs to ensure that these features are not correlated with protected attributes.

Lastly, we want to highlight that regardless of the form such data collection may take, it is imperative to consider the ethics of doing so and respect wishes and privacy of the individuals whose data are utilized during the model building process and who are affected by the model predictions.

## 7 Conclusion

This paper introduces a library of modular bias mitigators and ensembles and details experiments that confirm ensembles can improve fairness stability. We also provide generalizable guidance to practitioners based on their data setup.

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

## A Broader Impact Statement

After careful reflection, the authors have determined that this work presents no notable negative impacts to society or the environment.


## C  Supplemental Material

### C.1  Additional Tables

Table 6: Granular feature importance rankings of protected attributes for each dataset.

| Dataset | Protected Attribute Rankings |
|---|---|
| COMPAS Violent | sex: 4 
 race: 7 |
| Credit-g | age: 22 
 sex: 43 |
| COMPAS | sex: 5 
 race: 7 |
| Ricci | race: 6 |
| TAE | native_english_speaker: 1 |
| Titanic | sex: 2 |
| SpeedDating | importance_same_race: 24 
 samerace: 69 |
| Bank | age: 17 |
| MEPS 19 | RACE: 22 |
| MEPS 20 | RACE: 18 |
| Nursery | parents: 3 |
| MEPS 21 | RACE: 10 |
| Adult | sex: 19 
 race: 30 |

Table 7: Feature importance information for first-ranked feature for each dataset.

| Dataset | Feature 1 Name | Imp. |
|---|---|---|
| COMPAS V. | priors_count=More than 3 | 0.49 |
| Credit-g | checking_status_no checking | 0.11 |
| COMPAS | priors_count=More than 3 | 0.54 |
| Ricci | combine | 1 |
| TAE | native_english_speaker | 0.33 |
| Titanic | boat_13 | 0.79 |
| SpeedDating | like | 0.08 |
| Bank | poutcome_success | 0.18 |
| MEPS 19 | WLKLIM=2.0 | 0.34 |
| MEPS 20 | WLKLIM=2.0 | 0.26 |
| Nursery | health_not_recom | 0.59 |
| MEPS 21 | WLKLIM=2.0 | 0.19 |
| Adult | marital-status_Married-civ-spouse | 0.42 |

Table 8: Feature importance information for second-ranked feature for each dataset.

| Dataset | Feature 2 Name | Imp. |
|---|---|---|
| COMPAS V. | age_cat=Less than 25 | 0.22 |
| Credit-g | other_parties_guarantor | 0.03 |
| COMPAS | age_cat=Less than 25 | 0.25 |
| Ricci | position_Captain | 0 |
| TAE | summer_or_regular_semester_1 | 0.29 |
| Titanic | sex | 0.04 |
| SpeedDating | attractive_o | 0.05 |
| Bank | contact_unknown | 0.08 |
| MEPS 19 | ARTHDX=1.0 | 0.06 |
| MEPS 20 | ARTHDX=1.0 | 0.06 |
| Nursery | has_nurs_very_crit | 0.08 |
| MEPS 21 | ARTHDX=1.0 | 0.10 |
| Adult | education-num | 0.05 |

Table 9: Feature importance information for third-ranked feature for each dataset.

| Dataset | Feature 3 Name | Imp. |
|---|---|---|
| COMPAS V. | age_cat=Greater than 45 | 0.15 |
| Credit-g | credit_history_all paid | 0.03 |
| COMPAS | age_cat=Greater than 45 | 0.07 |
| Ricci | position_Lieutenant | 0 |
| TAE | course | 0.15 |
| Titanic | boat_A | 0.04 |
| SpeedDating | funny_o | 0.04 |
| Bank | month_mar | 0.05 |
| MEPS 19 | ACTLIM=1.0 | 0.03 |
| MEPS 20 | INSCOV=3.0 | 0.02 |
| Nursery | parents | 0.07 |
| MEPS 21 | ACTLIM=2.0 | 0.04 |
| Adult | capital-gain | 0.05 |

Table 10: Feature importance information for fourth-ranked feature for each dataset.

| Dataset | Feature 4 Name | Imp. |
|---|---|---|
| COMPAS V. | sex | 0.05 |
| Credit-g | savings_status_no known savings | 0.03 |
| COMPAS | priors_count=0 | 0.06 |
| Ricci | oral | 0 |
| TAE | course_instructor | 0.13 |
| Titanic | parch | 0.02 |
| SpeedDating | attractive_partner | 0.03 |
| Bank | month_jun | 0.04 |
| MEPS 19 | ADSMOK42=-1.0 | 0.02 |
| MEPS 20 | ACTLIM=1.0 | 0.02 |
| Nursery | has_nurs_critical | 0.06 |
| MEPS 21 | ACTLIM=1.0 | 0.03 |
| Adult | occupation_Other-service | 0.03 |

Table 11: Feature importance information for fifth-ranked feature for each dataset.

| | Feature 5 | |
|---|---|---|
| Dataset | Name | Imp. |
| COMPAS V. | age_cat=25 to 45 | 0.03 |
| Credit-g | property_magnitude_no known property | 0.03 |
| COMPAS | sex | 0.03 |
| Ricci | written | 0 |
| TAE | class_size | 0.11 |
| Titanic | body | 0.02 |
| SpeedDating | funny_partner | 0.03 |
| Bank | duration | 0.04 |
| MEPS 19 | JTPAIN=1.0 | 0.02 |
| MEPS 20 | ACTLIM=2.0 | 0.02 |
| Nursery | has_nurs_improper | 0.03 |
| MEPS 21 | INSCOV=3.0 | 0.02 |
| Adult | relationship_Own-child | 0.03 |

Table 12: Optimal pre-estimator mitigator configurations (with corresponding hyperparameters) per dataset. Hyperparameter names are not provided if the mitigation technique only accepts one. If a hyperparameter is not listed in the rightmost column, the configuration utilizes the default value.

| Dataset | Mitigator | Hyperparameters |
|---|---|---|
| COMPAS Violent | DisparateImpactRemover | 1 |
| Credit-g | LFR | k=5, Ax=0.01, Ay=10, Az=5 |
| COMPAS | DisparateImpactRemover | 0.4 |
| Ricci | LFR | k=5, Ax=0.01, Ay=5, Az=10 |
| TAE | LFR | k=5, Ax=0.01, Ay=50, Az=5 |
| Titanic | DisparateImpactRemover | 0.8 |
| SpeedDating | DisparateImpactRemover | 0.2 |
| Bank | DisparateImpactRemover | 0.2 |
| MEPS 19 | LFR | k=5, Ax0.01, Ay=1, Az=10 |
| MEPS 20 | LFR | k=5, Ax=0.01, Ay=1, Az=10 |
| Nursery | LFR | k=20, Ax=0.01, Ay=1, Az=10 |
| MEPS 21 | LFR | k=5, Ax=0.01, Ay=1, Az=10 |
| Adult | LFR | k=5, Ax=0.01, Ay=1, Az=10 |

Table 13: Optimal in-estimator mitigator configurations (with corresponding hyperparameters) per dataset. Hyperparameter names are not provided if the mitigation technique only accepts one. If a hyperparameter is not listed in the rightmost column, the configuration utilizes the default value.

| Dataset | Mitigator | Hyperparameters |
|---------|-----------|-----------------|
| COMPAS Violent | MetaFairClassifier | 0.5 |
| Credit-g | AdversarialDebiasing | classifier_num_hidden_units=10 |
| COMPAS | MetaFairClassifier | 0.5 |
| Ricci | MetaFairClassifier | 0.8 |
| TAE | MetaFairClassifier | 0.8 |
| Titanic | MetaFairClassifier | 1 |
| SpeedDating | MetaFairClassifier | 0.9 |
| Bank | PrejudiceRemover | 100 |
| MEPS 19 | PrejudiceRemover | 1000 |
| MEPS 20 | AdversarialDebiasing | classifier_num_hidden_units=500 |
| Nursery | MetaFairClassifier | 0.5 |
| MEPS 21 | AdversarialDebiasing | classifier_num_hidden_units=500 |
| Adult | PrejudiceRemover | 1000 |

