# OpenReview forum: "Searching for Fairer Machine Learning Ensembles"
_automl.cc/AutoML/2023/Conference — AutoML 2023 MainTrack_

### Review · Reproducibility_Reviewer_yqtn · 2023-04-06

**Completeness Of Code And Dataset Supplement Rating:** 3
**Usability And Ease Of Reproducibility Rating:** 3

**Actions Required To Increase The Reproducibility And Overall Recommendation:**

I would be convinced to increase my review rating given updates w.r.t. the following points (as mentioned above):

* Fixing the bugs I have encountered or clarifying what I might have done wrong,
* Improving the documentation and code structure,
* Adding (or pointing me to) the code of the missing components,
* and improving the submission checklist.

**Completeness Of Code And Dataset Supplement:**

Note, the following only discusses points mentioned above that are insufficient. All other points were found in the code supplements.

The `cluster_vis_preprocessing.py` has no executable version and is not properly documented in the README or code. I was not able to make it run properly and also did not see its connection to the paper. It also requires non-documented requirements.

The code for hyeropt requires non-documented requirements (`TensorFlow` or `lela[all]`).

I was not able to get the hyperopt search code to run. Initially, it failed due to a multiprocessing bug as the script is missing a `__name__ == "__main__"` guard. Then, with such a guard, I ran into a recursion depth problem or a crash without any exceptions. I have not found the reasons for this. My best guess is that it has to do with the imports, but I was not able to figure it out in a reasonable amount of time by debugging lela's code. For context, I tried this on a Windows machine and Linux machine, as no system requirements were provided in the readme or requirements.txt (nor the used Python version).

The hyperparameters for the ensemble methods seem to be limited to the number of estimators and default parameters. For these, the values described in the paper (Section 4) and in the code do not match. The code has more values for the grid search than mentioned in the paper. I am not sure what was used for the final results.

All grid search implementations failed for me. The `fit_model` function of each `run_{{ensemble_method_name}}_configuration.py` seems to be wrong, or the requirements are wrong. The score methods require `score_data` instead of `scoring`. Moreover, for VotingClassifer, `predict_proba` does not exist as you are using its default values. As far as I can tell, `voting` is set to `hard` and, hence, the model does not support `predict_proba`. After fixing this, I was able to make the grid search run, but not with all metrics for Voting.
In general, I was not able to replicate the results exactly, since the models' random seeds are missing (as mentioned in the submission checklists), but I think it is *very likely* that the saved results were created with this code.

I have not verified this for all possible combinations, but it seems that the shared result data is incomplete. For instance, I was not able to find the gird search results for RICCI for stacking. But I was able to produce my own results for this combination.

I was not able to find the code related to preprocessing the results (Section 5), the LOO evaluation (Section 5), or the code that computed the final result values for the Tables (e.g., Tables 2 and 3). I found code to generate plots that are not in the paper, and, as far as I can tell, do not directly represent something discussed in the paper. As none of these components is properly documented in the README or the code files, I might have missed this, and it is actually in any of these files. But I have not found it by looking through each file individually. Feel free to correct me, if I am mistaken.

**Overall Reproducibility Review:**

## Positive

The code and data supplements are almost complete. It seems to include most code required for reproducibility and for anyone else to work with and extend the work presented in this paper. Considering the amount of required code for and different components of this project, the authors' efforts to enable reproducibility were good.

The checklist is filled out in a sufficient way, as it answers most of the questions researchers might have about the paper.

## Negative

The code documentation and project structure can be improved a lot. The README barely covers all necessary components of the project and is very brief. Moreover, the project structure makes it hard to understand the code, and most in-code documentation is missing. As a result, the bugs make it harder to reproduce the results, and I would like to see a working step-by-step version of each component of the project.

The checklist requires minor adjustments to be fully representative of the paper. Currently, some of the answers could be more concrete to make it easier for a researcher to understand your work and underline its validity.

**Review Confidence:**

4: You are confident in your assessment, but not absolutely certain. It is unlikely, but not impossible, that you did not understand some parts of the submission or that you are unfamiliar with some pieces of the code or data.

**Review Rating:**

8: Accept, all aspects of this are reproducible with minor effort.

**Review Summary:**

Overall, the code supplements produced by the author are good but require some additional effort to be complete and easily usable by other researchers. However, bugs and missing documentation would unnecessarily stop other researchers from using the code and reduce the work's impact. The submission checklist can also be improved.

**Summary Of Necessary Code And Dataset Supplement:**

The authors' description and statements in the paper require the following code or dataset supplements:

- 8 mitigators (from AIF360)
- 4 ensemble methods (+ hyperparameters): bagging, boosting, voting, and stacking from sklearn
- usage of 13 datasets (from OpenML and other sources) with specific preprocessing
- build on the Lale library
- hyeropt with BO as a multi-objective optimizer for CASH
- custom gird search
- specific results preprocessing: mapping to the point of optimal fairness; min-max scaling of mean and  standard deviation
- specific result presentation: code to obtain values used in the plots, leave-one-out (LOO) evaluation
- shared their code via https://anonymous.4open.science/r/fairer_ensembles/voting_gridsearch.py (last updated by authors on March 27th 2023; last accessed 7th April 2023)


**Usability And Ease Of Reproducibility:**

As mentioned above, as part of "Completeness Of Code And Dataset Supplement", I believe the results are reproducible with some additional effort.

But, overall, the code quality and documentation can be improved. There seems to be a lot of potential for abstracting away code into function calls or loops. Moreover, I would recommend employing different directories for different parts of the project instead of a list of files.

The file structure and documented workflow are also not correct for the current project structure - the output and input paths of different files do not match or reference the actual output directory. Running the commands in order does not produce the intended output (assuming the bugs that I encountered, mentioned above, were resolved).

Finally, I would like to note, that the chosen method of sharing the code (https://anonymous.4open.science/) required me to manually download each individual file of your code. If the authors switch to AutoML's instance of anonymous GitHub (https://anon-github.automl.cc/) or share a ZIP file via OpenReview, this would not be necessary and could make the (updated) code much easier to use in the rebuttal period.

---

> ### Author Response · Authors · 2023-05-01
> **Author response to Reproducibility Reviewer yqtn**
>
> Thank you for your thorough and diligent reproducibility review!
>
> We have updated the code and requirements.txt for the reproduction package to include lale[full] and a compatible xgboost version. The installation and execution was tested on Mac OS version 12.6. Furthermore, as suggested by you, we have moved it to https://anon-github.automl.cc/r/fair_ensembles-BC93. Please note that the original experiments used a distributed Linux (RHEL) cluster, so we did not have the issue with multiprocessing for hyperopt_search.py. However, we have made changes so that the experiments can be reproduced on a single machine. Also, the results for RICCI with stacking were present in the original repo, but were somehow lost during the anonymization. The repo at the new location should contain all the raw results.
>
> Furthermore, we have updated the submission checklist based on your feedback, elaborating on several answers and revising a couple.

---

> > ### Comment · Reproducibility_Reviewer_yqtn · 2023-05-02
> > **Response: Change for Reproducibility**
> >
> > Dear Authors,
> >
> > Thank you for the changes and the additional effort put into making your work reproducible.
> >
> > I am convinced by the response, inspecting the updated code, and the changes to the checklist that the authors resolved my concerns. I have raised my score accordingly.

---

### Official Review · Reviewer_MqGH · 2023-04-11

**Potential Impact On The Field Of Automl Rating:** 3
**Technical Quality And Correctness Rating:** 3
**Clarity Rating:** 3
**Actions Required To Increase Overall Recommendation:** Change the captions of tables and als…

**Summary Of Contributions:**

In this study, authors explore the trade-off between the fairness and performance of different models through using 13 data-sets, eight mitigators and four ensembles. They also, suggest a guidance diagram to address this trade-off based on insights stemmed from this exploration.



**Clarity:**

The paper is well-written for the most part and it was easy to follow the concepts and methodology. However, I found some issues that I want to address:

1. For AutoML and some other conferences, table captions need to be above the table.
2. Some of the tables filled half of the width of the paper which I believe was because you wanted to fit more tables/figures in the table. Due to that, in some places the formatting of the text broke(e.g. page 8 before discussion).


**Overall Review:**

This study used a large number of data-sets and configurations to explore the trade-offs between fairness and performance which is an important topic in the community.

However, I have some minor issues that I did not quite figure out.

1. What was the reasoning behind 8000 rows/samples in guidance diagram? How would that be helpful or beneficial to take such split?

2. For Table 6, authors mention that "while the guidance diagram rarely recommends the exact same con- figuration as that found by Hyperopt, it recommends one with similar performance". However, roughly in half of cases(across all metrics) they have <0.1 differences and others are higher than that. Therefore, that claim is not valid. Also, is there any explanation why almost all cases of guidance F1's DI has >0.1 difference with Hyperopt suggestion(in some of them the difference is significant)?

**Potential Impact On The Field Of Automl:**

This paper aims to tackle the trade-off between the fairness and predictive performance which is ubiquitous kind of question in real-world challenges. I believe using such frameworks would be helpful in a lot of scenarios for AutoML community.

**Review Confidence:**

3: You are fairly confident in your assessment. It is possible that you did not understand some parts of the submission or that you are unfamiliar with some pieces of related work.

**Review Rating:**

6: Borderline Leaning Accept: Technically sound paper where reasons to accept outweigh reasons to reject. Please use sparingly.

**Review Summary:**

I believe this paper explores one of the most important trade-offs across classical machine learning that would be beneficial for AutoML community as well. With addressing the formatting problem mentioned in clarity section, I believe it has a chance to be accepted.

**Technical Quality And Correctness:**

I believe for the most part, this study was well conducted and I appreciate the authors endeavors.

---

> ### Author Response · Authors · 2023-05-01
> **Author response to Reviewer MqGH**
>
> Under "Actions Required To Increase Overall Recommendation", you requested that we "Change the captions of tables and also try to fit them properly." We have done both, and furthermore, we have replaced Table 6 by a new Figure 5 as suggested by Reviewer 4CRV.
>
> Under "Clarity", you asked "What was the reasoning behind 8000 rows/samples in guidance diagram?" This was motivated by the observation that volatility is highest in small datasets, and hence, the guidance for small datasets likely differs. You also remarked that the claim that the guidance diagram recommends configurations with similar performance is not valid. We have weakened that claim accordingly.

---

### Official Review · Reviewer_cvkx · 2023-04-11

**Potential Impact On The Field Of Automl Rating:** 3
**Technical Quality And Correctness Rating:** 3
**Clarity Rating:** 3

**Summary Of Contributions:**

The article discusses the challenge of maintaining algorithmic fairness in machine learning models when using bias mitigators, as their effectiveness can vary across different data splits. The authors explore the use of ensemble learning as a potential solution but note that it is unclear how to best combine ensembles with mitigators to balance fairness and predictive performance. They present an open-source library that allows for the modular composition of mitigators and ensembles along with hyperparameters, which they use to explore the configurations of 13 datasets. The authors provide their insights in the form of a guidance diagram that can serve as a starting point for practitioners. They also demonstrate that their solutions perform similarly to those generated by automatic combined algorithm selection and hyperparameter tuning (CASH) on many datasets.

**Actions Required To Increase Overall Recommendation:**

Please provide the time and memory usage when fitting models and discuss more about the extra overhead

**Clarity:**

The authors identify an important challenge in maintaining algorithmic fairness in machine learning models and propose an approach that combines ensemble learning with bias mitigators to balance fairness and predictive performance. They present an open-source library and a guidance diagram that can be useful for practitioners in the field of AutoML.

**Overall Review:**

Addressing an important challenge: The paper addresses an important challenge in maintaining algorithmic fairness in machine learning models, which is a crucial issue in the field of AI.

Open-source library: The paper presents an open-source library that enables the modular composition of mitigators and ensembles along with hyperparameters, which can be useful for practitioners in the field.

Empirical exploration: The authors explore the configurations of 13 datasets empirically, which can provide insights into the effectiveness of their approach.

Guidance diagram: The authors provide their insights in the form of a guidance diagram that can serve as a starting point for practitioners, which can be helpful for those who are new to the field of AutoML.

this work adopts grid search to find the best mitigation configs, which would be inefficient when the search space is large

**Potential Impact On The Field Of Automl:**

This paper has potential importance for the field of AutoML as it addresses an important challenge in maintaining algorithmic fairness in machine learning models. The open-source library presented in the paper enables the modular composition of mitigators and ensembles, which can be useful for practitioners in the field. The insights distilled from the exploration of configurations on 13 datasets in the form of a guidance diagram can also serve as a starting point for practitioners.

As such, it is likely that this paper will be cited by researchers and practitioners in the field of AutoML who are interested in improving the fairness of machine learning models. However, the extent of its citation will depend on how widely the open-source library is adopted and how effective the guidance diagram proves to be in practice.

**Review Confidence:**

2: You are willing to defend your assessment, but it is quite likely that you did not understand the central parts of the submission or that you are unfamiliar with some pieces of related work.

**Review Rating:**

6: Borderline Leaning Accept: Technically sound paper where reasons to accept outweigh reasons to reject. Please use sparingly.

**Review Summary:**

The open-source library and guidance diagram provided by the authors may be useful for practitioners in the field of AutoML who are interested in improving the fairness and accuracy of their machine learning models.

**Technical Quality And Correctness:**

They used both brute-force and Hyperopt search to optimize fairness while maintaining decent predictive performance. It's not clear about the efficiency of the proposed library although the authors mentioned that they measure time and memory but didn't provide them.

---

> ### Author Response · Authors · 2023-05-01
> **Author response to Reviewer cvkx**
>
> Under "Actions Required To Increase Overall Recommendation", you wrote "Please provide the time and memory usage when fitting models and discuss more about the extra overhead". In response, we have added a new Figure 3, along with a new paragraph entitled "How do ensembles affect resource consumption?", in the results section.

---

### Official Review · Reviewer_y2RT · 2023-04-12

**Potential Impact On The Field Of Automl Rating:** 2
**Technical Quality And Correctness Rating:** 3
**Clarity Rating:** 3

**Summary Of Contributions:**

The paper explores the combination of multiple bias mitigators and ensembles to improve algorithmic fairness and predictive performance. The authors developed an open-source library and guidance diagram for the composition of mitigators, ensembles, and hyperparameters, and conducted experiments with the configuration space on various datasets.

**Actions Required To Increase Overall Recommendation:**

highlight the importance and potential impact of this work for the field of AutoML

**Clarity:**

Although the paper is well structured, for readers who are not well-versed in the terminology of "fairness in machine learning", it may prove challenging to comprehend. Thus, key terms such as the different ensemble techniques, bias mitigators, or group fairness matrices should be introduced early in the paper. Additionally, I recommend to enhance the formatting of the tables, as they presently appear to be floating amidst the text.

**Overall Review:**

The paper demonstrates technical rigor and holds the potential to have impact for "fairness in machine learning" while incorporating some elements of AutoML. For readers unfamiliar with the terminology, the paper may be challenging to comprehend. It is recommended to introduce key terms early in the paper, improve the formatting of tables to enhance readability, and compare with relevant existing research.

**Potential Impact On The Field Of Automl:**

Instead of using an ensemble-specific bias mitigator for one specific kind of ensemble, 8 off-the-shelf mitigators and 4 off-the-shelf ensemble techniques have been explored. Based on this, the authors contribute a guidance diagram. Despite the inclusion of AutoML elements (i.e. automated search via Bayesian optimization), the main emphasis of this work lies in fairness, resulting in a limited contribution to the field of AutoML. In essence, the work appears to primarily utilize AutoML techniques to advance research in a different domain.

**Review Confidence:**

2: You are willing to defend your assessment, but it is quite likely that you did not understand the central parts of the submission or that you are unfamiliar with some pieces of related work.

**Review Rating:**

5: Borderline Leaning Reject: Technically sound paper where reasons to reject nonetheless outweigh reasons to accept. Please use sparingly.

**Review Summary:**

To me it is uncertain whether the AutoML Conference is a suitable venue for publishing this work.

**Technical Quality And Correctness:**

Exploring the Cartesian product of datasets, ensemble techniques, mitigators, and hyperparameters through both manual grid search and automated search using Bayesian optimization is a reasonable approach. Nonetheless, the proposed method lacks comparison with relevant existing research.

---

> ### Author Response · Authors · 2023-05-01
> **Author response to Reviewer y2RT**
>
> Under "Actions Required To Increase Overall Recommendation", you requested that we "highlight the importance and potential impact of this work for the field of AutoML". It is widely accepted that ensembling (often post-hoc, but not always) is a critical part of AutoML (see for example AutoSklearn, AutoGluon). The motivation behind this wide use of ensembling is that it often improves the predictive performance over the base estimators. In our paper, we systematically try to understand the influence of ensembling on both predictive performance and fairness. Our elaborate empirical study allows us to tease out of the effects of the different ensembling and mitigation choices. This is critical if we wish to systematically incorporate bias mitigation in AutoML systems.  Furthermore, before our paper, it was unknown where bias mitigation should be introduced in an ensemble.  The search space we use pushes the boundaries of what is possible with current AutoML search tools, since it involves multiple layers of nested meta-estimators, while requiring the search to consider multiple options both inside and outside of these layers. Our paper presents novel search spaces and our implementation allows us to search over these novel search spaces. We have updated the related-work section with citations for ensembles in AutoML.
>
> Thank you for your feedback towards improving the clarity of the paper. We have expanded Section 3 (library and datasets) to define disparate impact and provide more background on ensembles. Furthermore, we have reformatted the tables to make them easier to read.

---

### Official Review · Reviewer_4CRV · 2023-04-16

**Potential Impact On The Field Of Automl Rating:** 3
**Technical Quality And Correctness Rating:** 4
**Clarity Rating:** 3
**Ethics And Accessibility Rating:** Yes, regarding other reasons (please …

**Summary Of Contributions:**

The paper combines bias mitigators and ensembling methods, and shows that ensembling reduces the volatility of the mitigators. They present a flow chart for choosing a bias mitigator and ensemble combination to use.

**Actions Required To Increase Overall Recommendation:**

Show results presented as tables as figures, to make them easier to interpret, and show that the results compared to random guessing and the best-performing combination.

**Clarity:**

Clarity
The main point is that the results in the tables, especially Table 6, are hard to interpret, given the amount of numbers. Can you present this in a plot? Additionally:
- Could you include a definition of disparate impact, and short descriptions of the mitigators you compare?
- Bold table values: what values are bolded, and did you check whether there was a significant difference, e.g. between 0.08 and 0.09 in No Mit. DV Table 2.
- Can you walk through the algorithm in Figure 2?
- Can you compare the performance from the guidance diagram to the best choice and a random choice?
- What would you suggest for future work?

Minor:
- What do you mean by worldview (l40)?
- Table 6 caption, line 4 has a trailing sentence.

Typos:
- L85: comma should be a full stop.


**Ethics Details (Optional):**

Accessibility: The blue and green in table 6 is hard to make out. Can you change it to bold and underlined or something similar?

**Overall Review:**

The paper provides multiple contributions: a flow chart for choosing ensemble/mitigator combinations, empirical evidence for the ensembles helping, a library for combining ensembles and mitigators, and some analysis of the quality of the flow chart. The results could be made more interpretable by presenting them as plots, but overall I think it is a useful paper.

**Potential Impact On The Field Of Automl:**

This can help make bias mitigators more accessible, especially given the open-source library.

**Reproducibility (Optional):**

I disagree with the response of 'yes' to 3J given that they do not report random seeds.

**Review Confidence:**

2: You are willing to defend your assessment, but it is quite likely that you did not understand the central parts of the submission or that you are unfamiliar with some pieces of related work.

**Review Rating:**

8: Accept: Technically sound paper with major impact and strong evaluation, with perhaps some minor flaws.

**Review Summary:**

The paper compares a number of bias mitigators and ensembling methods, and probably most usefully provides a flow chart with suggestions for what combination to use. This could be very useful, though the results could be presented in a more easy-to-digest manner.

**Technical Quality And Correctness:**

There are parts where the technical quality was hard to assess due to how results were presented, see ‘Clarity’.

---

> ### Author Response · Authors · 2023-05-01
> **Author response to Reviewer 4CRV**
>
> Thank you for your positive review! Based on your recommendation, we have replaced Table 6 by a new Figure 5, which we hope is easier to read and more accessible. We have also made other changes throughput the paper based on the feedback in your review.

---

> > ### Comment · Reviewer_4CRV · 2023-05-02
> > **Thank you for your response**
> >
> > Thank you for your changes.
> >
> > I keep my postitive score.